# Wilson’s Disease: An Update on the Diagnostic Workup and Management

**DOI:** 10.3390/jcm10215097

**Published:** 2021-10-30

**Authors:** Beata Kasztelan-Szczerbinska, Halina Cichoz-Lach

**Affiliations:** Department of Gastroenterology with Endoscopy Unit, Medical University of Lublin, 8 Jaczewski Street, 20-954 Lublin, Poland; halina.lach@umlub.pl

**Keywords:** anti-copper therapy, ATP7B, copper chelators, liver disease, Wilson’s disease, zinc salts

## Abstract

Wilson’s disease (WD) is a rare autosomal recessive disorder of hepatocellular copper deposition. The diagnostic approach to patients with WD may be challenging and is based on a complex set of clinical findings that derive from patient history, physical examination, as well as laboratory and imaging testing. No single examination can unequivocally confirm or exclude the disease. Timely identification of signs and symptoms using novel biomarkers and modern diagnostic tools may help to reduce treatment delays and improve patient prognosis. The proper way of approaching WD management includes, firstly, early diagnosis and prompt treatment introduction; secondly, careful and lifelong monitoring of patient compliance and strict adherence to the treatment; and, last but not least, screening for adverse effects and evaluation of treatment efficacy. Liver transplantation is performed in about 5% of WD patients who present with acute liver failure at first disease presentation or with signs of decompensation in the course of liver cirrhosis. Increasing awareness of this rare inherited disease among health professionals, emphasizing their training to consider early signs and symptoms of the illness, and strict monitoring are vital strategies for the patient safety and efficacy of WD therapy.

## 1. Introduction

WD was firstly described more than 100 years ago by Samuel Kinnier Wilson [1]. Excessive copper deposition in the body organs, particularly in the liver and brain, is a typical feature of the disease which results from a mutation in the copper-transporting gene ATP7B. Predominant patterns of WD presentation include hepatic, neurologic, and psychiatric disorders, although diverse signs and symptoms of the disease have been described so far [2,3]. This rare genetic disorder requires prompt diagnosis and lifelong treatment, and both significantly influence patient prognosis. We aimed to summarize current knowledge on WD concerning its pathogenesis, clinical manifestations, and diagnostic workup, as well as to address current WD management with some intended future treatment possibilities.

## 2. Etiology and Epidemiology of the Disease

WD is a rare autosomal recessive inherited disorder of hepatocellular copper deposition. Mutations of the P-type ATPase copper transporter gene, ATP7B, which was identified in human chromosome 13, are responsible for the development of the WD signs and symptoms. ATP7A and ATP7B regulate cytoplasmic copper by pumping it out of cells or into the endomembrane system [4]. ATP7B null mutations induce swelling of mitochondria in the liver and nervous system, with a collection of these engorged structures in the perikaryon. To date, over 600 variations in ATP7B have been described, including the most common single-nucleotide missense and nonsense mutations, as well as insertions and deletions, and rare splice site mutations [5]. The most common are His1069Glu (H1069Q) found in Europe and North America; Arg778Leu in South Korea, Japan, and China; 2007del7 in Iceland; and Met645Arg in Spain. Most of the patients are heterozygotes, i.e., they have two different mutations, therefore genetic diagnosis in the disease is not easy and usually takes several months. The highest incidence of Wilson’s disease in the world was reported in Costa Rica (4.9 per 100,000 inhabitants) with Asn1270Ser mutation and in Sardinia (1/10,000−1/7000) with the (−441/−427del) mutation. In Europe, the disease is most frequently diagnosed in Austria (3.0 per 100,000 inhabitants) and Germany (2.5 per 100,000 inhabitants) [6,7]. Copper is an important component of various vital enzymes in the metabolic process, e.g., lysyl oxidase, cytochromeC oxidase, and superoxide dismutase. [8] The average daily consumption of copper in humans is 2–5 mg depending on the individual consumption of meat, chocolate, and seafood, and it exceeds the metabolic requirement of the human body [9]. About 50–75% of the ingested copper is absorbed from the intestinal lumen and passes to the liver; this process is normal in patients with Wilson’s disease. Inside the liver cells, copper is incorporated into several enzymes and proteins binding this element, including apoceruloplasmin, which, as a result, is converted to ceruloplasmin (plasma ferroxidase belonging to alpha2-globulin). When the level of copper in the liver exceeds safe levels, the P-type ATPase exerts dual action both in the Golgi apparatus and hepatocyte cytoplasm. ATP7B loads copper molecules onto apoceruloplasmin and enables their further secretion into the plasma. In the cytoplasm, ATP7B separates excess copper into vesicles, from where it is excreted via exocytosis through the apical cellular membrane into the bile [5]. However, when an abnormal form of ATPase is produced as a result of a mutation of the ATP7B gene, the process of copper incorporation into ceruloplasmin, as well as the excretion of excess copper into the bile, is impaired [10]. Ceruloplasmin is a serum copper-binding protein and its levels are low in patients with WD. Excessive accumulation of copper into hepatocytes leads to subsequent cell damage and massive copper release into the bloodstream. Therefore, free unbound serum copper levels are elevated in WD. From the blood, it further passes to other tissues and organs. A high copper concentration in WD is observed mainly in the brain, eye corneas, and kidneys. In patients with WD, the copper balance in the body is influenced by impaired bile excretion but not by its intestinal absorption, which is normally regulated [11]. It seems that rather than the excessive copper collection in tissues, the free blood fraction (5–15%) is responsible for the toxicity in patients with WD. Unbound copper acts as a potent oxidant and induces the production of highly reactive hydroxyl radicals, resulting in further lipid peroxidation of cell membranes, DNA, protein, and mitochondria damage [12]. The toxic effects of the unbound copper are also connected to copper-induced dysfunction of apoptosis inhibitors and the loss of control over caspase-3 [11,13].

## 3. Clinical Presentation of Wilson’s Disease

Wilson’s disease can appear at any age, although it is most prevalent in patients before the age of 40. Most cases are diagnosed between 5 and 35 years of age [3]. Clinical signs and symptoms of the disease may vary considerably but the most common are hepatic (including cirrhosis), neurologic, and psychiatric disorders; ophthalmic signs (Kayser–Fleischer rings); and episodes of hemolysis coexisting with acute liver failure [14]. Due to its heterogeneous presentation, Wilson’s disease has been referred to as “the great masquerader” [15].

### 3.1. The Hepatic Alterations in Wilson’s Disease

The hepatic symptoms of Wilson’s disease may precede the onset of neuropsychiatric disorders by up to 10 years. About 50% of patients with WD present with liver disease. Hepatic steatosis is the earliest histological feature in WD and may reflect copper-induced impairment in the function of mitochondria [16,17]. It may be indistinguishable from non-alcoholic fatty liver disease (NAFLD). Mitochondrial changes, increased peroxisomes’ fat droplets, lipolysosomes, and intranuclear glycogen inclusions have been described as the most frequent hepatic ultrastructural changes in WD patients [18]. As the copper content of hepatic tissue is not measured in every case, electron microscopic examination seems to be a valuable diagnostic tool for the early detection of the disease. Moreover, ultrastructural evaluation may be helpful to distinguish between heterozygous carriers and WD patients [19]. Liver histology may have several different patterns, i.e., mild non-specific changes, steatosis or steatohepatitis, chronic hepatitis, and acute hepatitis with submassive or massive necrosis [20]. None of these are specific to WD. Similarly, variable clinical manifestations of liver disease have been described in WD patients. They include (1) clinically asymptomatic liver disease that can only be confirmed by biochemical abnormalities, imaging, or histopathological changes, which still occurs only in rare WD cases as even a liver biopsy may not show any organ alterations; (2) chronic hepatitis; (3) cirrhosis of the liver (compensated or decompensated), which is the most common liver disorder at WD diagnosis; and (4) acute liver failure, which usually occurs in young women (assessed female to male ratio is 4:1) or in previously diagnosed patients who have stopped their treatment [20,21,22]. The acute liver failure in the course of Wilson’s disease should be suspected in every patient with intense jaundice and only a slight elevation of the activity of transaminases, low alkaline phosphatase, and cholinesterase levels, and low hemoglobin concentrations may occur. Occasionally, the condition may be associated with Coombs-negative hemolytic anemia and acute renal failure. Moreover, the retrospective multicenter cohort study of 1186 patients revealed that the rate of hepatobiliary malignancies in WD is very low, even in cirrhotic patients [23]. Similar results were obtained from a Dutch retrospective cohort study in 130 patients with WD who were followed during a median follow-up of 15 years (range of 0.1–51.2). Although data do not support regular HCC surveillance in patients with WD [24], the risk of carcinogenesis is increased in these patients. HCC occurrence has been reported both in cirrhotic [25,26] and non-cirrhotic WD patients [27]. Furthermore, some reports highlight the need for consideration of liver cancer development, even in young patients with WD [28]. Moreover, the American and European Association for the Study of Liver Diseases (AASL and EASL) recommend HCC screening in patients with liver cirrhosis regardless of etiology [29,30].

### 3.2. The Neurologic Changes in Wilson’s Disease

Neurological disorders are the second most frequent clinical symptoms of WD after liver disease. They may appear early in the course of this disease, through parallel hepatic disorders, or several years after its onset. Prominent neuropsychiatric symptoms can confuse the initial diagnosis and delay treatment [14,15]. However, most patients with neurologic manifestations are already diagnosed with liver cirrhosis. The basal ganglia and brainstem are regions with the highest susceptibility to copper toxicity and their injury leads to various combinations of movement and psychiatric disorders [17]. The severity of neurologic disorders in the course of WD may differ from subtle symptoms recurring from time to time for several years to rapid and acute disorders that lead to complete disability in a short time. The most common early signs of WD include ataxia, clumsiness of the face, dysarthria, hypersalivation, and personality changes. Late manifestations, which are less prevalent with appropriate WD treatment, include dystonia, tremors, parkinsonism, seizures, and choreoathetosis [14,31]. Based on the most common neurologic disorders, WD is classified into four types of clinical syndromes: (1) Parkinson-like syndrome (45%) with face masking, speaking problems, involuntary hand movements, and muscle stiffness, and, in particular, the diagnosis of the so-called “Juvenile Parkinsonism” should lead to WD suspicion; (2) multiple sclerosis-like syndrome (pseudosclerosis) (24%), with predominantly tremors; (3) ataxic syndrome (15%), wherein excessive muscle tension with abnormal limb movements dominate; and (4) chore-like syndrome (11%), with predominantly abnormal movements and dystonia [14]. However, a large number of patients present more than one type of neurologic disorder, therefore sometimes it is difficult to categorize them unambiguously into one of the aforementioned subgroups. Additionally, in patients with end-stage liver disease, the neurologic manifestations of WD may be incorrectly recognized as hepatic encephalopathy.

### 3.3. The Psychiatric Manifestation of Wilson’s Disease

At the time of diagnosis, about 10 to 20% of patients with WD present psychiatric disorders [32]. They mainly include emotional lability, increased impulsivity, sexual exhibitionism, and self-harm tendencies. More than one psychiatric disorder may be present. Depression is quite common and worthy of mentioning [32]. Children may present with isolated psychiatric symptoms as the first WD sign [33]. Sometimes, these disorders may be mistaken for adolescence-period problems. Regarding neurologic signs, psychiatric disorders are classified into four groups: (1) behavioral disorders; (2) affective disorders; (3) schizophrenia-like type disorders; and (4) cognitive disorders.

### 3.4. The Ophthalmic Signs and Symptoms in the Course of Wilson’s Disease

Kayser–Fleischer rings, which correspond to copper deposits in the descement membrane around the cornea, are observed in about 95% of WD patients presenting with neurologic signs and symptoms, and may be absent in up to 50% of both WD patients with isolated hepatic disease and most asymptomatic individuals. Moreover, Kayser–Fleisher rings may be observed in other liver diseases, e.g., in primary biliary cholangitis (PBC) [34]. The rings can be visualized in a non-invasive ophthalmological examination using a slit lamp [35]. Kayser–Fleischer rings are considered to be relevant criteria for confirmation of Wilson’s disease [36]. Furthermore, their disappearance can be observed as a result of favorable therapy. Patients with Wilson’s disease may present with another eye disorder called the sunflower cataract. The special pattern of copper deposition in the middle point of the lens forms a disc with radial strands that resemble a sunflower, hence the name of the disorder [37]. Reports on the incidence of sunflower cataract (SC) in WD patients are limited and inconsistent. Results of two small ophthalmological studies, in which 52 and 53 heterogeneous patients with WD were assessed, confirmed SC presence in 1.9 and 17% of patients [34]. However, a more recent study carried out in Polish patients found SC in only one (1.2%) of the newly diagnosed WD individuals. It completely disappeared after one year of treatment for WD. The authors concluded that SC might be a rare and reversible ophthalmological manifestation of WD and observed only at the time of the disease diagnosis, disappearing with chelation therapy [38].

### 3.5. The Hematologic Alterations in Wilson’s Disease

Acute hemolytic anemia may occur as a rare complication of Wilson’s disease [39]. It appears in about 10–15% of WD cases. In most patients, it is Coombs-negative and occurs as a result of toxic intravascular damage to erythrocytes caused by high blood copper concentrations. This type of anemia may be associated with acute liver failure. Therefore, any patient with acute hepatic failure and Coombs-negative hemolytic anemia, together with low levels of transaminases and alkaline phosphatase, as well as an alkaline phosphatase to bilirubin ratio of below 2, should be checked for Wilson’s disease. Other hematologic alterations, including thrombocytopenia and leukopenia, may result from hypersplenism due to liver cirrhosis or may be side effects of WD therapy [3,22].

### 3.6. The Renal Manifestation of Wilson’s Disease

The incidence of renal complications in Wilson’s disease varies greatly. The expression of the gene causing Wilson’s disease has been confirmed in the kidneys, therefore dysfunction may be a primary or secondary phenomenon in the release of copper from the liver [40]. Damage to the renal tubules caused by copper deposits leads to symptoms that may resemble Fanconi syndrome and may be associated with both the presence of renal acidosis and excessive excretion of amino acids, glucose, fructose, galactose, uric acid, phosphorus, and calcium in the urine. Approximately 16% of patients have urolithiasis, which is usually related to excessive excretion of calcium in the urine and the impaired acidification process. Hematuria and nephrolithiasis were also observed in the course of the disease. Proteinuria in this disease may appear isolated or secondary as a side effect of D-penicillamine treatment [2].

### 3.7. The Bone–Muscular Alterations of Wilson’s Disease

Symptoms and signs of the joint disease occur in 20–50% of patients with Wilson’s disease in its late stage, most often after the age of 20, and may resemble osteoarthritis [40]. Arthropathies and joint pains usually affect the joints of the spine and large joints of the limbs, wrists, knees, and hips. Aseptic arthritis, chondromalacia, and chondrocalcinosis have also been described, as well as a case of acute rhabdomyolysis. Osteopenia is found in more than half of the patients in radiographs. Osteoporosis, rickets, and spontaneous bone fractures have also been reported in the course of the disease [14]. Sometimes, the clinical and radiological symptoms of the osteoarticular disease are the first harbingers of metabolic disorders in Wilson’s disease and lead to its diagnosis.

### 3.8. Symptoms and Signs from Other Systems and Tissues in the Course of Wilson’s Disease

In Wilson’s disease, cardiovascular involvement can be also observed as arrhythmias, cardiac myopathy, and small vessel disease. Recent reports have confirmed that signs and symptoms associated with cardiovascular complications are frequent indications for hospitalization in WD patients [41,42]. Additionally, altered skin pigmentation and a bluish color at the base of the nails (so-called azure nipples) have been described. Dermatoses may follow therapy, e.g., with D-penicillamine [40].

## 4. Diagnostic Approach to Patients with Wilson’s Disease Suspicion

Early WD diagnosis is essential to prevent further disease complications. The diagnostic approach is based on a complex set of clinical findings that derive from patient history, physical examination, laboratory, and imaging diagnostic testing. The isolated genetic assessment may be insufficient because genetic variation does not always impact protein functionality [43,44]. In patients with neurologic and/or hepatic disorders suggesting the disease, evaluation towards the presence of ophthalmic signs, i.e., Kayser–Fleischer rings, and determination of the blood level of ceruloplasmin (levels lower than 10 mg/dL at the normal range of 20–50 mg/dL) should be performed. In clinically asymptomatic patients with no Kayser–Fleischer rings who present with isolated liver test abnormalities, liver biopsy is required (copper content in the liver tissue above 250 mg/g of dry weight confirms the diagnosis), as well as the determination of the ceruloplasmin level in the blood. In clinically asymptomatic patients with no Kayser–Fleischer rings who present with isolated liver test abnormalities, a liver biopsy may be considered (copper content in the liver tissue above 250 mg/g of dry weight confirms the diagnosis). However, liver dry copper estimation has some limitations, i.e., high copper concentration in liver samples was observed in biliary atresia. Moreover, uneven hepatic distribution of the element and its low tissue level in WD patients with advanced liver disease may cause biopsy sampling errors with false-negative results [45]. Currently, hepatic copper quantification has been removed from the modified Leipzig scoring system [46]. Determination of the blood ceruloplasmin level, urinary copper excretion, and molecular analysis generally is sufficient to confirm the diagnosis. Genetic verification is currently also more accessible for a larger number of patients. Therefore, invasive tests such as liver biopsy should be avoided according to new modified guidelines [46].

### 4.1. Biochemical Evaluation for the Diagnosis of Wilson’s Disease

#### 4.1.1. Serum Ceruloplasmin

A low level of ceruloplasmin in the blood, usually below 0.2 g/L (normal range: 0.2–0.5 g/L) or below 50% of the lower limit of the normal range, may be indicative of WD [3,10,34]. However, since ceruloplasmin belongs to the acute phase proteins, its level may increase to normal values in the coexistence of inflammatory diseases and induce false-negative results. A similar situation occurs in hyperestrogenemia during pregnancy or estrogen therapy. Additionally, its low values may be seen in other liver diseases (i.e., autoimmune hepatitis, severe liver failure of a different etiology, and familial aceruloplasminemia) as well as in conditions leading to renal or intestinal protein loss; in malabsorption disorders, e.g., celiac disease; and in heterozygous carriers of ATP7B mutations who do not show copper accumulation in their body [10,14,34].

#### 4.1.2. Urinary Copper

As a rule, high 24 h urinary copper excretion, which usually exceeds 100 µg/24 h in adults and 40 µg/24h in children with WD, confirms the presence of the illness. The test remains the most sensitive single screening tool in the diagnosis of WD [34]. Usually, in patients with neuropsychiatric forms of the disease, urinary copper excretion increases to more than 100 µg/24 h (normal range: 0–50 µg/24 h). In asymptomatic heterozygotes, elevated urinary copper levels are sometimes observed but usually they do not exceed 40 µg/day. Additionally, in severe liver failure, a false-positive high copper excretion can be found. After the re-evaluation of the penicillamine challenge test in children, it was found unreliable with too-low sensitivity to rule out WD in asymptomatic children [47]. Due to the ambiguity of the results obtained from clinical trials conducted in adults with WD, the test is not recommended for them either [34]. Determination of urinary copper excretion is also an important aspect of controlling the proper treatment of WD patients [14,46,48].

#### 4.1.3. Serum Copper

Total serum copper (TSC) consists of copper incorporated in ceruloplasmin and non-ceruloplasmin-bound copper called free copper. The TSC diagnostic value is poor as it does not indicate tissue concentrations. Since 90% of the copper in the blood is bound to ceruloplasmin, the determination of its free (i.e., non-ceruloplasmin-bound NCC), toxic, and high susceptibility to tissue deposition fraction would be more adequate for the WD diagnosis. NCC can be calculated as follows: NCC μg/dL = total serum copper level (μg/dL)-3 x ceruloplasmin level (mg/L) [10,14]. Nevertheless, NCC measurements are method-dependent, related to the accuracy of serum copper and ceruloplasmin tests [34,48], and have been proven unreliable [49]. In 2009, a new method for the direct determination of labile copper, called the exchangeable copper (CuEXC) fraction, was developed and evaluated as a diagnostic tool for WD [50]. CuEXC provides data on the free-copper overload as well as on the spread and severity of WD [51]. Moreover, relative exchangeable copper (REC = CuEXC/total copper%) has been already proven to be an excellent biomarker for WD diagnosis with 100% sensitivity and 100% specificity [52].

#### 4.1.4. Coombs-Negative Hemolytic Anemia

Coombs-negative hemolytic anemia in patients with WD may occur in up to 15% of individuals [53]. It appears to be a specific laboratory finding associated with acute liver failure (ALF) due to WD and seems to be related to increased oxidative stress induced by copper accumulation within red blood cells [3,54].

#### 4.1.5. Blood Liver Tests

Hepatic dysfunction is the typical feature of WD seen in more than 50% of patients [34]. The condition should be considered in the differential diagnosis of any unexplained abnormal liver tests, especially in patients below the age of 40. The laboratory alterations may vary from asymptomatic abnormal liver tests, chronic hepatitis, and liver cirrhosis to acute liver failure. Liver chemistries may manifest as a hepatocellular injury with liver enzyme alanine transaminase (ALT) and aspartate transaminase (AST) elevation, and an increased bilirubin level. Decreased albumin concentration and coagulopathy with prothrombin time prolongation or thrombocytopenia (due to hypersplenism and portal hypertension) usually occur when the end-stage liver disease develops [3]. Previous studies demonstrated that findings such as low values of serum alkaline phosphatase and increased aspartate aminotransferase:alanine aminotransferase (AST:ALT) ratios were associated with fulminant WD [55]. Furthermore, the combination of a alkaline phosphatase (ALP) to total bilirubin (TB) ratio below 4 and an AST to ALT ratio above 2.2 yields 100% diagnostic sensitivity and specificity in predicting acute liver failure (ALF) due to WD (WD-ALF) [56]. Careful monitoring of liver enzyme levels is warranted in all WD patients.

Taking into account a wide variety of clinical circumstances, a complex panel of diagnostic tests should be performed when WD is suspected. No single examination can unequivocally confirm or exclude the disease.

#### 4.1.6. ATP7B Protein Quantification: A New Non-Invasive Diagnostic Assay

Quantification of the ATP7B protein in dried blood may serve as an adjunctive test for the diagnosis of WD. The measurements were made by immunoaffinity enrichment mass spectrometry. As the results of the study show, the test effectively identified WD patients in the group of the 92.1% of studied cases and reduced ambiguities resulting from ceruloplasmin and genetic analysis [57].

### 4.2. Genetic Testing for At-Risk-of Wilson’s Disease Individuals

Molecular and genetic diagnostics in WD are not easy due to numerous and rare mutations, and due to the heterozygosity of the majority of patients, as mentioned earlier [3,5]. For this reason, the correct diagnosis is usually made after several months. Failure to confirm the mutation does not exclude WD [5]. However, the guidelines indicate the validity of testing for the ATP7B gene mutation in each patient with WD suspicion in order to confirm the diagnosis and to initiate the screening of first and second-degree relatives of a given patient [34]. Generally, the lack of an evident genotype–phenotype association in the disease course indicates a need for further investigation of alternative mechanisms, such as modifier genes and the epigenetic regulation of gene expression, to explain the pleiotropic effects of WD [58]. It seems that as advanced molecular techniques develop, recommendations in this regard will change.

### 4.3. Radiologic Imaging in Wilson’s Disease

Neuroimaging should be performed in all WD patients with or without neurological symptoms. Brain magnetic resonance imaging (MRI), which is the most sensitive imaging tool for the diagnosis of WD, has been included in the modified Leipzig scoring system [46]. The procedure is used for the confirmation of cerebral copper accumulation and basal nuclei damage. Patients with neurological WD signs and symptoms almost always present with magnetic resonance imaging (MRI) brain alterations. T1-weighted MRI predominantly detects atrophic changes while T2-weighted MRI records signal changes in the putamen [59]. The “face of the giant panda” sign present in the midbrain is considered a characteristic WD feature [60]. Results of Polish neuroimaging studies of WD patients confirmed MRI brain abnormalities in the neuropsychiatric, hepatic, and presymptomatic WD individuals in 90.4%, 41.7%, and 25% of the cases, respectively. Moreover, some brain MRI gender differences were described [61]. Recently, a semiquantitative scale for cerebral MRI abnormalities has been validated and proven reliable, as well as a valid instrument for the assessment of WD severity [62]. Additionally, a relatively new technique called quantitative susceptibility mapping (QSM) has been assessed and reported as effective in indicating increased magnetic susceptibility changes in the basal ganglia and brain stem of WD patients, as well as in detecting them before any evident alterations on T_1_ and T_2_-weighted MR images appear [63]. Several other new imaging methods have been assessed recently for WD diagnosis. Anterior segment optical coherence tomography can be used for the detection and quantification of Kayser–Fleischer rings as the better option compared to a slit-lamp examination [64]. Dynamic PET analysis with copper-64 chloride has been assessed in the WD mice model for functional imaging of copper metabolism imbalance in WD [65]. MR spectroscopy (MRS) of the brain has been evaluated for the detection of early neurological changes related to WD. The prospective pediatric study indicates that MRS detects metabolite abnormalities before cerebral structural changes occur in MRI, but it also may confirm early neurological changes even with normal MRI. MRS is also helpful for monitoring disease progression [66].

### 4.4. Histopathological Examination in Wilson’s Disease

Liver biopsy in WD is indicated only when the clinical picture and/or the results of non-invasive tests do not allow for confirmation of the final diagnosis of the disease and/or there is a suspicion of another coexisting disease in the liver [18]. Staining with rhodamine or orcein reveals focal hepatic copper concentration in less than 10% of patients because these dye agents detect lysosomal deposits of the chemical element. Therefore, the method of choice for the diagnosis of WD is an examination of the quantitative copper content in the liver parenchyma [20]. The length of the liver sample submitted for the examination should be at least 1 cm and does need to be frozen. Copper concentration over 4 μmol/g of dry mass is the best biochemical indicator of WD. Due to the uneven distribution of copper in the liver parenchyma in the advanced stages of the disease, the possibility of sample error should be taken into account. Therefore, normal hepatic copper content does not rule out WD diagnosis [18]. Additionally, in humans over 6 months of age, a positive result of staining of the hepatic tissue for the presence of copper occurs almost exclusively in liver diseases such as Wilson’s disease but also in chronic cholestatic diseases, including PBC, PSC, liver cirrhosis, and primary liver tumors (most often fibrolamellar hepatocellular carcinoma) [14,20].

### 4.5. Criteria for the Diagnosis of Wilson’s Disease

The Leipzig criteria were developed to assist and standardize WD diagnosis and management [34]. A score of 4 and more points confirms WD. Unfortunately, the assessment based on the Leipzig scoring system does not always provide evident conclusions [67,68] and although genetic testing is currently more available, some patients still cannot obtain it. The criteria for WD diagnosis included in the Leipzig score are presented in Table 1.

In 2019, the new modified Leipzig scoring system for the diagnosis of WD was recommended by Nagral et al [46]. Additional points were added for typical brain changes seen on MRI and positive family history with deaths due to hepatic or neurological complications were made suggestive for WD.

* if no quantitative liver copper is available. ULN, an upper limit of normal. Adapted from the EASL Clinical Practice Guidelines: Wilson’s disease. *J. Hepatol.*
**2012**, 56, 671–685 [34].

## 5. Management of Wilson’s Disease

The proper way of approaching WD management includes the following steps: firstly, early diagnosis and prompt treatment introduction; secondly, careful and lifelong monitoring of patient compliance and strict adherence to the treatment, and, last but not least, screening for adverse effects and evaluation of treatment efficacy [34,69].

### 5.1. Dietary Recommendations

The negative copper balance remains a relevant aim of patient management. Patients with WD should avoid copper-rich meals containing liver, chocolate, nuts, and seafood (especially lobster), and should consume low-copper water [70]. These recommendations are of particular importance in the first year of treatment. Current evidence indicates that in stable WD patients who are adherent to medical therapy dietary, copper restrictions may be unnecessary with two food exceptions (shellfish and liver). Alcohol abstinence and avoidance of hepatotoxic drugs remain permanent recommendations for life. A recent report indicates that a high-calorie diet severely aggravates hepatic mitochondrial and hepatocellular damage in the animal model of Wilson’s disease [71].

### 5.2. Pharmacotherapy

In 2012, the recommendations of the European Association for the Study of the Liver (EASL) regarding WD therapy were published [34]. The experts emphasize that the treatment of WD should be lifelong and discontinuation of the therapy may lead to serious disease consequences, including death in a short time. The only rationale for discontinuing therapy may be liver transplantation. Since biochemical defects of WD are located inside hepatocytes, liver transplantation leads disease curing [14]. Anti-copper treatment for symptomatic WD consists of two periods: an initial period with active copper removal and a second maintenance period wherein the disease is well controlled and the dosage should be decreased to avoid copper deficiency. The current therapy for WD is presented in Table 2 and Table 3.

Contemporary guidelines recommend the administration of chelating agents (d-penicillamine and trientine) as first-line therapy for symptomatic WD patients. The latter medication seems to be better tolerated. Both agents promote urine copper excretion [34,76].

#### 5.2.1. D-Penicillamine

Its mechanism of action includes the advancement of the urinary excretion of copper but also quite possibly the induction of endogenous metallothionine [12]. D-penicillamine should be administered in two to four doses of 1.5–1 g/day an hour before or 2 h after a meal. Since the risk of neurologic deterioration is high, D-penicillamine should be initiated gradually and slowly, especially in the subgroup of patients with typical brain alterations, even if they present with no neurologic signs and symptoms [77]. The maintenance dose after achieving remission is 0.75–1 g/day. In patients with a negative copper balance, the lowest effective dose should be used. For elderly patients and children below 12 years of age, 20 mg/kg of body weight/day is recommended in divided doses. The doses for children above 12 years old are the same as for adults. In patients with liver dysfunction, clinical improvement usually occurs within the first 2–6 months of treatment, but complete remission can be achieved one year after the beginning of therapy. In patients with neurologic disorders, WD signs and symptoms disappear more slowly and remission may require up to 3 years. D-penicillamine as a pyridoxine antagonist increases the vitamin excretion in urine. Therefore, it is recommended to supplement 25–50 mg/day of vitamin B6 in order to prevent anemia and/or inflammation of peripheral nerves that may occur as a result of pyridoxine deficiency. Furthermore, D-penicillamine influences collagen metabolism and may inhibit liver fibrosis; additionally, it may delay wound healing. The use of D-penicillamine is limited by its side effects that include early allergic reactions, lymphadenopathy, leukopenia, and thrombocytopenia as a result of bone marrow suppression, myasthenic syndrome, lupus-like syndrome, kidney dysfunction, and dermatological changes, which may require discontinuation and modification of treatment in about 30% of patients [69]. The adverse effects of D-penicillamine usually disappear with time after switching to trientine and do not recur with this type of treatment.

#### 5.2.2. Trientine

Trientine’s mechanism of action is similar to D-penicillamine in that it promotes urinary excretion of copper. Trientine should be titrated up to 1–2 g/day divided into two to four doses. Iron supplementation is contraindicated during trientine therapy as toxic complexes are produced as a result of copper–iron chelates. Potential side effects of trientine include pancytopenia, hemorrhagic gastritis, loss of taste, systemic lupus erythematosus, and neurologic deterioration (according to Brewer et al. about 26% of WD patients experience this symptom during the initial treatment) [46,78]. Since food reduces the absorption of chelating agents by up to 50%, they should be administered 1 h before or 2 h after a meal. Based on literature reports, chelating drug treatment should be initiated slowly, i.e., at the beginning, ¼– ½ of the target dose should be administered and then should be titrated by half a tablet every 4–7 days to reach the whole recommended dose within 1–2 weeks [46,76]. However, there is no definitive protocol on the rate of dose escalation.

Careful assessment of the biochemical response as well as clinical monitoring are vital for patient safety. The aforementioned dosage strategies and the patient approach help to avoid the neurologic deterioration after the sudden hepatic copper mobilization (it may occur in 10–50% of those treated with D-penicillamine and in less than 26% of patients treated with trientine) [78,79,80]. After remission is reached, the dose of chelating agents can be reduced to 1 g/day divided into two daily doses for maintenance therapy.

#### 5.2.3. Tetrathiomolybdate

Ammonium tetrathiomolybdate (TTM), a strong de-coppering medication, is not commercially available due to the limited experience in its administration. Several previous clinical trials with this preparation were not successful probably due to its instability [19,81]. Nevertheless, a new therapeutic option is bis-choline tetrathiomolybdate (TTM), which undergoes current examination. Its mechanism of action includes the formation of stable copper–TTM–albumin complexes, inhibition of liver and neuronal copper uptake, and promotion of copper biliary excretion. Recently, an open-label, multicenter phase 2 study confirmed the efficacy of bis-choline TTM in 28 patients with WD [74]. The new TTM complex demonstrated to be more stable than that formed by the previous TTM compound and had better bioavailability. Moreover, bis-choline TTM presents the ability to pass through the blood–brain barrier and further into neuronal cells. Since negatively charged chelators cannot cross the blood–brain barrier, the management of neurologic disorders is the main challenge in WD patients. Therefore, bis-choline TTM may become a novel treatment strategy useful in neurologic-predominant WD [19,74].

#### 5.2.4. Zinc Salts

The efficacy and safety of zinc salts for WD treatment have been confirmed in clinical studies. They are increasingly and continuously being used as the first-line treatment for initial and maintenance therapy in WD patients [45]. Zinc salts seem to be as effective as D-penicillamine in both asymptomatic and symptomatic WD patients but are better tolerated. Neither therapy appears to be superior [82,83]. One recent systematic review, which included 23 studies with 2055 patients’ evaluations, showed that zinc salts are safer than D-penicillamine therapy and similarly effective in preventing or ameliorating both hepatic and neurological WD symptoms. However, due to low study quality, cautious interpretation of the results is warranted [84].

Zinc salts’ major advantage is an extremely low level of toxicity, except for dyspepsia [84,85,86]. Currently, oral zinc preparations are recommended as the first-line treatment in patients with the neurologic forms of WD, since, as mentioned before, signs and symptoms from the nervous system may worsen as a result of chelation therapy [78]. Zinc is also safe in the treatment of pregnant women (category A according to the Food and Drug Administration classification). Zinc’s mechanism of action differs from those of chelating agents. It includes stimulation of the metallotheionine (MT) synthesis (an endogenous protein metal chelator) in the epithelium of intestinal mucosa [87,88]. This protein preferentially binds copper in enterocytes and inhibits its permeation into the portal circulation. Rapid desquamation of epithelial cells leads to a negative copper balance. By causing a negative copper balance, zinc salts mobilize the release of the element from the body reserves. The alternative effect of zinc action is liver metallothionein induction to bind copper and prevent toxic hepatocyte damage [89]. Various zinc salts are used in WD treatment, including sulfate, acetate, and gluconate. They do not differ in their effectiveness but may differ in their tolerance mainly due to their different capability to gastric mucosa irritation. The adverse effect may occur in approximately 10% of patients and tends to decrease over time. A recent study by Antczak-Kowalska M. et al. indicates that patients receiving zinc sulfate had increased gastropathy risk compared to those receiving no treatment or D-penicillamine [90]. The EASL guidelines for WD treatment do not provide any recommendation on what type of zinc salt to administer. Zinc acetate is the only zinc salt currently approved by the U. S. Food and Drug Administration [34,91]. In Poland, zinc sulfate is most commonly administered. The recommended dose is 150 mg of elemental zinc per day (75 mg for children weighing less than 50 kg), administered 30 min before a meal in three divided doses. It is still not known whether combination therapy with zinc and chelators is more effective than monotherapy (there are no objective clinical trials available). Certainly, both drug groups should be given at different time intervals to prevent chelators from neutralizing the effects of zinc salts [69]. During zinc salt administration, close monitoring of the liver function is required. If transaminase levels increase, copper chelators should be initiated. A copper chelator or a zinc preparation may be used as maintenance therapy for asymptomatic patients or those with neurologic signs and symptoms.

Despite the AASLD [92], EASL [34], and ESPGHAN pediatric [48] guidelines, there are no results from prospective randomized trials comparing the effectiveness of individual medications used in WD treatment thus far. These guidelines are the current best practice for the management of WD.

## 6. Future Perspectives in Wilson’s Disease Treatment

Attempts to use antioxidants (vitamin E and curcumin) in WD therapy are ongoing but results and further recommendations are not yet available [93,94,95,96]. ATP7B mutations, including the two most frequent variants, i.e., H1069Q and R778L, cause protein product collection in the endoplasmic reticulum. As a result, they cannot arrive at copper excretion sites, leading to the toxic accumulation of the chemical element in the liver. Modification of the location of the aforementioned mutants and their movement to the right cell functional area might re-establish copper excretion in WD patients [97]. Nevertheless, the determination of molecular targets for the adjustment of endoplasmic reticulum-retained ATP7B mutants remains a demanding job. Recently, Chesiet al. reported that p38 and c-Jun N-terminal kinase might be such targets for the correction of WD-causing mutants and, hence, could be evaluated for the development of novel therapeutic strategies to combat WD [98]. Larger studies are required to confirm the results. Several new therapeutic options for WD are currently under evaluation in clinical trials including gene cell therapy. So far, only bis-choline tetrathiomolybdate (TTM and ALXN1840, formerly WTX101) assessed in the FoCus Phase III trial seems to demonstrate a significant impact in mobilizing intracellular copper in the liver. It is the first-in-class medication that reduces plasma non-ceruloplasmin-bound copper (NCC) and, as the targeted de-coppering medication, selectively binds to as well as removes copper from both the blood and tissues [74]. The Alexion Clinical Phase III Trial is now recruiting WD patients in the United States and Europe to investigate the long-term efficacy and safety of ALXN1840 in an up to 60-month period. The other two clinical trials in gene therapy are also recruiting WD patients both in the United States and Europe. The first one, the GATEWAY trial, will explore VTX-801, which is a corrective version of the ATP7B gene packaged within a vector engineered to deliver it into liver cells. Additionally, studies performed in animal models indicate that adeno-associated virus (AAV) gene therapy based on truncated ATP7B is a promising strategy in the treatment of WD [99]. The next trial, called the CYPRUS2+ study, will assess UX701, which is a gene therapy product intended to correct the defect of ATP7B [100].

Cell therapy in WD is currently also under evaluation [75,101]. Recent reports indicate that fulminant hepatitis in WD may be prevented by hepatocyte transplantation. The procedure increased short-term survival in an animal model [102]. The successful transplantation of healthy hepatocytes into the rat liver emphasizes the potential of future cell therapy in humans with WD.

## 7. Liver Transplantation for Wilson’s Disease

Liver transplantation (LT) is performed in about 5% of WD patients who present with acute liver failure at first disease presentation or with signs of decompensation in the course of liver cirrhosis [103,104]. Copper metabolism is normalized after LT and anti-copper medications are no longer required. Although LT remains a curative treatment and restores the normal function of hepatocytes, LT is not recommended solely for neurologic or psychiatric WD variants. Nevertheless, there are some reports stating that neurologic disorders resolve with such treatment. In contrast, there are also reports of severe neurologic deterioration in WD patients despite successful liver transplantation [105]. The King’s College Modified Scale criteria, as developed by Dhawan et al., are used for patient assessment and a score of 11 or more indicates a high risk of death as well as requires a prompt liver transplant [106]. There are few reports concerning extracorporeal albumin dialysis (MARS) as a bridge to liver transplantation in WD patients [107,108].

## 8. Pregnancy and Wilson’s Disease

Pregnancies of women with WD may constitute some problems, although a positive outcome of both the mother and newborn can be achieved upon principles of close monitoring and adequate treatment. Several reports indicate that the procreative success in women with WD is directly related to treatment efficacy [109]. Since the risk of having homozygous children was estimated at 0.5%, the determination of the haplotype of a WD patient’s partner is also recommended. Treatment of females suffering from WD should be continued throughout pregnancy. D-penicillamine (FDA category C) at a dose of 0.75–1.5 g/day seems to be safe. There is a certain risk of teratogenicity with both chelators and zinc salts, but the risk of discontinuing therapy is far greater than the risk of continuing therapy [109]. Some experts recommend reducing the dose of d-penicillamine and trientine in the first trimester of pregnancy with close monitoring of patients. Others indicate that the doses of copper chelators should be reduced to 300–600 mg/day in the last trimester of pregnancy to ensure an adequate supply of copper to the fetus and to avoid disturbances in wound healing if termination of pregnancy by cesarean section is required [110,111]. Breastfeeding is not recommended during chelation therapy. Contraception is a relevant issue for women with WD. So far, no studies have been conducted, allowing for objective recommendations in this matter. There is some evidence that estrogen may interfere with the biliary excretion of copper. Moreover, intrauterine devices contain copper. For these reasons, spermicides, barrier agents, and preparations containing only progesterone are the best contraceptive options for women with WD [34].

## 9. Long-Term Monitoring of Patients with Wilson’s Disease

Patients suffering from WD should be monitored throughout their life in order to supervise the effects of therapy and the progression of both disease signs and symptoms so the treatment may be modified as soon as possible [112]. As reported recently, not taking anti-copper treatment regularly had an important negative effect on patient clinical outcomes [113]. The patient evaluation frequency is determined individually based on the time elapsed since the treatment initiation, the overall treatment effectiveness, a need for drug conversion, as well as a degree of patient cooperation and compliance. In most cases, WD patients should be monitored every 2 months during the first year after the treatment initiation and one to two times a year thereafter [47,48,72]. Routine evaluations include clinical assessment, i.e., physical and neurologic examination, as well as laboratory tests that involve the determination of blood copper and ceruloplasmin levels; the grade of hepatitis (based on transaminase activity); liver cell synthetic function (based on INR, albumin, and urea levels); renal parameters; complete blood counts; and urinalysis. Patients with Kayser–Fleischer rings should be referred to an ophthalmologist for an eye examination in the light of a slit lamp once a year to confirm ring reduction with adequate copper elimination [34,40]. The evaluation of treatment quality is based on the determination of the daily urine copper excretion during both the treatment and two days after its discontinuation. This assessment should be carried out at least once a year. The values of copper excretion at the beginning of therapy should remain in the range of 500–1000 µg/24 h for D-penicillamine and 300–1000 µg/24 h for trientine, and during treatment maintenance, they should remain between 200 and 500 µg/24 h for both aforementioned medications. However, in patients treated with zinc salts, these values should remain below 100 µg/24 h and 30–80 µg/24 h depending on the treatment period. Moreover, a daily urine zinc excretion of more than 1.5–2.0 g/24 h confirms patient compliance. Another parameter helpful in assessing the effectiveness of therapy is non-ceruloplasmin-bound or free-copper blood concentration. A level above 25 µg/dL at the beginning of therapy and 15–25 µg/dL during maintenance treatment should be maintained [47,48,69,72].

An important aspect of therapy control in WD patients is the recognition of signs of overtreatment. These may include neutropenia, anemia, high transaminase, and high ferritin blood levels. In such cases, the free-copper concentration usually remains below 15 µg/dL and its urinary excretion is lower than in the patient’s pre-symptom period. Copper–chelator treatment should be temporarily discontinued or changed to zinc salts under close patient monitoring and then restarted with the lower dose [14,69].

## 10. Conclusions

Heterogeneous clinical presentations of WD due to copper deposition in different tissues and organs remain a diagnostic challenge. Early recognition of the signs and symptoms of the disease based on novel biomarkers and modern diagnostic tools may help to reduce treatment delays and improve patient prognosis. Increasing awareness of this rare inherited disease among health professionals, emphasizing their training to consider early WD symptoms, and strict patient monitoring are vital strategies for the patient safety and efficacy of WD therapy.

## Figures and Tables

**Table 1 jcm-10-05097-t001:** Diagnostic scoring system for Wilson’s disease.

Clinical and Laboratory Presentation	Points
**Kayser–Fleischer rings**	
Present	2
Absent	0
**Neurologic symptoms or typical abnormalities of brain MRI**	
Severe	2
Mild	1
Absent	0
**Serum ceruloplasmin (g/L)**	
Normal (>0.2)	0
0.1–0.2	1
<0.1	2
**Coombs-negative hemolytic anemia**	
Present	1
Absent	0
**Liver copper (in the absence of cholestasis)**	
>5 × ULN (>4 µmol/g)	2
0.8–4 µmol/g	1
Normal (<0.8 µmol/g)	−1
Rhodanine-positive granules *	1
**24 h urinary copper (in the absence of acute hepatitis)**	
Normal	0
1–2 × ULN	1
>2 × ULN	2
Normal but >5 × ULN after D-penicillamine	2
**Mutation analysis**	
Mutations detected on both chromosomes	4
Mutations detected on one chromosome	1
Mutations absent	0
**TOTAL SCORE:**	
Diagnosis established	4 or more
Diagnosis possible, more tests needed	3
Diagnosis very unlikely	2 or less

**Table 2 jcm-10-05097-t002:** Characteristics of medications available for the treatment of Wilson’s disease [45,46,48,72,73,74,75].

Medication	Treatment Indications	Adverse Effects	Potency and Efficacy
Agents mobilizing copper from tissues and increasing its urinary excretion (chelators)			
D -Penicillamine	First-line induction treatmen: the first oral chelating agent for WD treatment	Allergic reactions (fever and rash), lymphadenopathy, bone marrow suppression, lymphadenopathy, lupus-like syndrome, kidney dysfunction, and deterioration in neurological status	Very effective
Trientine	Second-line induction treatment: approved for patients intolerant of d-penicillamine	Autoimmune reactions, kidney dysfunction, bone marrow suppression, and deterioration in neurological status	Effective
Agents preventing copper absorption			
Zinc salts	Maintenance treatment: first-line induction treatment in selected patient subgroups (neurologic WD variant, intolerant to chelators, pregnant women, and asymptomatic WD patients)	Stomach irritation and otherwise a low level of toxicity	Effective
Agents forming copper–albumin complexes (currently under evaluation)			
Bis-choline tetra-thiomolybdate (TTM)	Planned for first-line induction treatment	Hepatitis and bone marrow suppression	Under assessment, very effective

**Table 3 jcm-10-05097-t003:** Adult and child dosages of currently available treatments for Wilson’s disease [45,46,48,72,73,74,75].

Group of WD Patients	Treatment Indications	Medication *	Recommended Dosage for Adults	Recommended Dosage for Children
Symptomatic WD patients	Initial treatment (6 to 12 months)	D–penicillamine	250 mg at alternating days, gradually increasing by 250 mg every 2–4 weeks until 1.0–1.5 g/day in two or three doses (no definitive protocol on the rate of dose escalation)	150–300 mg, titrated until 20 mg/kg/day, given in two or three doses; young adults should take 1.0 g (maximum 1.5 g) daily in two to four doses
		Trientine (heat-sensitive, stored at 2–8 °C)	750 mg, 1.5 g/day in three doses	20 mg/kg/day in two to three divided doses; young adults should take 1.0 g (max. 1.5 g) daily in two to three doses
		Zinc salts	150 mg of elemental zinc/day in three doses	Age > 16 years and body weight > 50 kg: 150 mg of elemental zinc/day in three doses Age 6–16 years and body weight < 50 kg: 75 mg of elemental zinc/day in three doses Age < 6 years: 50 mg of elemental zinc/day in two doses
	Maintenance treatment (lifelong therapy)	D–penicillamine	10–20 mg/kg/day and up to 0.75–1.0 g/day in two doses	900–1500 mg per day in two or three doses
		Trientine (heat-sensitive, stored at 2–8 °C)	900mg, 1.5 g/day (or 10–15 mg/kg/day) in two to three doses; one daily dose of trientine as a maintenance therapy has been suggested and is currently under evaluation	900 mg, 1.5 g/day (or 10–15 mg/kg/day) in two to three doses
		Zinc salts (treatment of choice)	Dosage presented above and tailored individually	Dosage presented above and tailored individually
Asymptomatic WD patients		Zinc salts (treatment of choice)	Dosage presented above and tailored individually	Dosage presented above and tailored individually
		D-penicillamine or trientine	Reduced dosage of 10–15 mg/kg in two to four dosages	

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
