# Peer review of "Wilson’s Disease: An Update on the Diagnostic Workup and Management"

_jcm, 2021, doi:10.3390/jcm10215097_

Round 1
Reviewer 1 Report
- This is a well-organized review with a crucial focus on the current shortcomings in our diagnostic evaluation of WD
- Last sentence in section 3.1. “Therefore, data do not support regular HCC surveillance in patients with WD.” Would consider re-wording this statement to be more clear (cirrhotic vs. non-cirrhotic) and less definitive. Although the findings from reference 23 are acknowledged the Diagnosis, Staging, and Management of Hepatocellular Carcinoma: 2018 Practice Guidance by the American Association for the Study of Liver Diseases (Marrero et al.) still recommends HCC screening in patients with cirrhosis (regardless of etiology). To imply to the reader that HCC surveillance is not warranted in this population is somewhat misleading.
- Section 6. Future perspectives in Wilson’s disease treatment. Consider adding references regarding ongoing trials in gene therapy in US & Europe.
- Section 9 Long-term monitoring of patients with Wilson’s disease. The last 5 sentences of this section suggest several goal levels of urinary copper excretion. Consider adding reference here .
Author Response
Response to the First Reviewer:
We would like to thank the Reviewer very much for taking the time to assess our manuscript and all the valuable comments that helped us to correct and improve our paper. We have addressed all the concerns raised and here is a point-by-point response to the reviewer’s comments and concerns.
Section 3.1.
Our response: This part of our manuscript has been corrected according to the Reviewer’s suggestion as follows:
Although data do not support regular HCC surveillance in patients with WD, the risk of carcinogenesis is increased in these patients. HCC occurrence has been reported in cirrhotic [25] as well as in non-cirrhotic WD patients [26 ] Furthermore, some reports highlight the need for consideration of liver cancer development even in young patients with WD [27]. Moreover, the American and European Association for the Study of Liver Diseases (AASL and EASL) recommend HCC screening in patients with liver cirrhosis regardless of etiology [28, 29]
Section 6.
Our response: new pieces of information have been added according to the Reviewer’s suggestions as follows:
Several new therapeutic options for WD are currently under evaluation in clinical trials including gene cell therapy. So far, only bis-choline tetrathiomolybdate (TTM, ALXN1840, formerly WTX101) assessed in the FoCus Phase III trial seems to demonstrate a significant impact in mobilizing intracellular copper in the liver. It is the first-in-class medication that reduces plasma non-ceruloplasmin-bound copper (NCC) and as the targeted de-coppering medication selectively binds to and removes copper from both the blood and tissues. [74]
The Alexion Clinical Phase III Trial is now recruiting WD patients in the United States and Europe to investigate the long-term efficacy and safety of ALXN1840 in an up to 60-month period.
The other two clinical trials in gene therapy are also recruiting WD patients both in the United States and Europe. The first one, the GATEWAY trial will explore VTX-801 a corrective version of the ATP7B gene packaged within a vector engineered to deliver it into liver cells. Also, studies performed in animal models indicate that adeno-associated virus (AAV) gene therapy based on truncated ATP7B is a promising strategy in the treatment of WD [100; 101].
The next trial called the CYPRUS2+ study will assess UX701 that is a gene therapy product intended to correct the defect of ATP7B. [102]
Cell therapy in WD is currently also under evaluation [75; 103]. Recent reports indicate that fulminant hepatitis in WD may be prevented by hepatocyte transplantation. The procedure increased short- term survival in an animal model [104] The successful transplantation of healthy hepatocytes into the rat liver raises the chance of future cell therapy in humans with WD.
Section 9.
Our response: References have been added according to the Reviewer’s suggestion.
Reviewer 2 Report
In their paper, the authors examine the presentation of Wilson's disease, the diagnostic workup, and the therapeutic management. The article is fairly interesting although the bibliography could be revised with more recent articles and new aspects of the disease, for example new therapies under evaluation and diagnostic tools could be more discussed. About children diagnosis and management, the only published pediatric guidelines (J Pediatr Gastroenterol Nutr. 2018 Feb;66:334-344) should be cited.
Some major conceptual and minor issues need to be clarified or rewieved:
- In Etiology and Epidemiology the abbreviation WD is redefined
- “Copper is an important component of various enzymes vital in the metabolic process” should be changed to “Copper is an important component of various vital enzymes in the metabolic process”
- In etiology and epidemiology, the authors state “the P-type ATPase in the Golgi apparatus loads copper onto apoceruloplasmin, so it can be eliminated in the bile [5].” This is not quite correct. Copper binds to apoceruloplasmin and is secreted in the plasma, while ATP7B directly expels excess copper through the apical cellular membran into the bile (Ref 5).
- “In patients with WD, the proper balance of copper in the body is maintained predominantly by regulating its excretion (95% in bile), and not its intestinal absorption.” Could the author better specify this sentence? In WD intestinal absorption is normally regulated while biliary copper excretion is impaired. The sentence can create misunderstandings.
- In paragraph 3.1 The hepatic alterations in Wilson's disease, in the sentence “Mitochondrial changes, increased peroxisomes fat droplets, lipolysosomes, and intranuclear glycogen inclusions have been described as the most frequent hepatic ultrastructural changes in WD carriers [18].” , did the authors mean heterozygous carriers or WD patients?
- 17% of WD patients with sunflower cataracts sounds very high. What is the reference? Ref 29 doesn’t indicate the percentage of affected patients.
- In paragraph 4. Diagnostic approach to patients with Wilson's disease suspicion, the authors affirm: “In clinically asymptomatic patients, with no Kayser- Fleischer rings, who present with isolated liver test abnormalities, liver biopsy is required (copper content in the liver tissue above 250 mg/g dry weight confirms the diagnosis), as well as the determination of the ceruloplasmin level in the blood.” Indeed, ceruloplasmin, urinary copper excretion and molecular analysis may be sufficient to confirm the diagnosis, avoiding invasive tests such as liver biopsy according to available guidelines.
- In paragraph 4.1 the authors explain the criteria for the diagnosis of WD. Perhaps, the title should be changed. The reader understands that the reported tests have to be evaluated for WD diagnosis, while the Cu / Cp ratio and free blood copper are not criteria for diagnosis. In addition, the determination of intrahepatic copper and genetic tests are required to confirm the diagnosis, but not included in the section. Furthermore, there is no mention about hemolytic anemia or typical neuropsychiatric symptoms as diagnostic tools. Regarding high level of free copper in the blood, this is not considered diagnostic criteria in Leipzig score until now; this test is not universally available and the indirect calculation of free toxic copper is not a reliable method. For these reasons it should mentioned, but not included in the section on diagnostic criteria of Wilson’s disease as the authors state.
- In Radiologic Imaging in Wilson’s disease, the authors affirm “radiological imaging are not useful for WD diagnosis”. This sentence is not exactly correct if you consider that the brain MRI is included in the Leipzig score. The sentence should be rephrased. Moreover, could the authors explain how second-level imaging might be useful in presymptomatic patients? Usually, they are prescribed for clinically evident neurological signs and it is difficult to use them as a screening tool.
- “The adverse effects of D-penicillamine disappear with time after switching to trientine and do not recur with this type of treatment.” The word “usually” should be added if the authors think about dermatologic effects such as progeric changes in the skin and elastosis perforans serpingosa that may persist after drug discontinuation.
- “Potential side effects of trientine include pancytopenia, hemorrhagic gastritis, loss of taste, systemic lupus erythematosus, and neurologic disorders (in less than 26% of patients)”. Please add the reference.
- “Chelating drug treatment should be initiated slowly i.e. at the beginning ¼- ½ of the target dose and then titrating it by half a tablet every 4-7 days to reach the whole recommended dose within 1-2 weeks”. Is it a suggestion based on the authors’ experience or based on references? It should be clarified.
- “Zinc salts are not as effective as copper chelators”. The authors support this sentence with three refrences. It should be noted that, of the cited articles, only Weiss et al. suggested the ineffectiveness of zinc treatment differently from Brewer et al. Furthermore, the authors of the cited metanalysis clearly state that zinc is as effective as penicillamine with fewer side effects even if the evaluation is based on low quality available articles. The sentence should be accordingly modified.
- “Despite the AASLD [57] and EASL guidelines [37], there are no results from prospective randomized trials comparing the effectiveness of individual medications used in WD treatment so far. These guidelines are the current best practice for the management of WD.” Perhaps, the ESPGHAN pediatric guidelines (JPGN 2018;66: 334–344) and the Indian guidelines (Journal of Clinical and Experimental Hepatology | January/February 2019 | Vol. 9 | No. 1 | 74–98) should be cited as other available, more recent, WD guidelines.
- Reference 21 is duplicated with reference 31.
- WD treatment doses for adults and children should be better indicated in the table
Author Response
We have attached the file with our response
